# QTL Mapping for Leaf Rust Resistance in a Common Wheat Recombinant Inbred Line Population of Doumai/Shi4185

**DOI:** 10.3390/plants14193113

**Published:** 2025-10-09

**Authors:** Yamei Wang, Wenjing Li, Rui Wang, Nannan Zhao, Xinye Zhang, Shu Zhu, Jindong Liu

**Affiliations:** 1College of Life Sciences, Langfang Normal University, Langfang 065000, China; wangyamei@lfnu.edu.cn (Y.W.); liwenjing@lfnu.edu.cn (W.L.); wangrui@lfnu.edu.cn (R.W.); hbndnannan@163.com (N.Z.); zhigancao@126.com (X.Z.); 2Institute of Crop Sciences, Chinese Academy of Agricultural Sciences, Beijing 100081, China

**Keywords:** common wheat, kompetitive allele-specific PCR (KASP), leaf rust, marker-assisted selection (MAS), quantitative trait locus (QTL)

## Abstract

Leaf rust, a devastating fungal disease caused by *Puccinia triticina* (Pt), severely impacts wheat quality and yield. Identifying genetic loci for wheat leaf rust resistance, developing molecular markers, and breeding resistant varieties is the most environmentally friendly and economical strategy for disease control. This study utilized a recombinant inbred line (RIL) population of Doumai and Shi4185, combined with the wheat 90 K single nucleotide polymorphisms (SNPs) chip data and maximum disease severity (MDS) of leaf rust from four environments, to identify adult plant resistance (APR) loci through linkage mapping. Additionally, kompetitive allele-specific PCR (KASP) markers suitable for breeding were developed, and genetic effects were validated in a natural population. In this study, 5 quantitative trait loci (QTL) on chromosomes 1B (2), 2A and 7B (2) were identified through inclusive composite interval mapping, and named as *QLr.lfnu-1BL1*, *QLr.lfnu-1BL2*, *QLr.lfnu-2AL*, *QLr.lfnu-7BL1* and *QLr.lfnu-7BL2*, respectively, explaining 4.54–8.91% of the phenotypic variances. The resistance alleles of *QLr.lfnu-1BL1* and *QLr.lfnu-1BL2* originated from Doumai, while the resistance alleles of *QLr.lfnu-2AL*, *QLr.lfnu-7BL1* and *QLr.lfnu-7BL2* came from Shi4185. Among these, *QLr.lfnu-1BL2*, *QLr.lfnu-7BL1* and *QLr.lfnu-7BL2* overlapped with previously reported loci, whereas *QLr.lfnu-1BL1* and *QLr.lfnu-2AL* are likely to be novel. Two KASP markers, *QLr.lfnu-2AL* and *QLr.lfnu-7BL*, were significantly associated with leaf rust resistance in a diverse panel of 150 wheat varieties mainly from China. Totally, 34 potential candidate genes encoded the NLR proteins, receptor-like kinases, signaling kinases and transcription factors were selected as candidate genes for the resistance loci. These findings will provide stable QTL, available breeding KASP markers and candidate genes, and will accelerate the progresses of wheat leaf rust resistance improvement through marker-assisted selection breeding.

## 1. Introduction

Wheat production is severely threatened by leaf rust, a devastating fungal disease caused by *Puccinia triticina* (Pt). This pathogen can bring yield losses exceeding 50% [1]. More seriously, global warming has driven an increasing trend in leaf rust epidemics across major wheat-growing areas worldwide, including Southern United States, South America, Canada, Eastern Europe, Egypt, and notably throughout China [2,3]. Severe yield losses due to leaf rust have occurred in 2008, 2009, 2012, 2015, and 2023 in China [4,5,6]. The annual wheat planting area in China covers approximately 23.7 million hectares, with leaf rust affecting around 15 million hectares each year, leading to an estimated yield loss of 3 million tons annual, particularly in the major wheat-producing Yellow and Huai River Valleys Wheat Zone. Hence, effective control of leaf rust is crucial for ensuring grain security.

Developing resistant varieties is the most economical and effective strategy to control leaf rust. Wheat relies on two main types of resistance: race-specific and race non-specific. Race-specific resistance, often called all-stage resistance (ASR), is usually controlled by one or a few major genes. It functions based on the gene-for-gene principle. This means resistance requires a specific interaction between a host resistance (*R*) gene and a corresponding pathogen avirulence (*Avr*) gene [7,8]. ASR is only effective against pathogen races that carry those specific *Avr* genes. However, ASR can be overcome as Pt races evolve new virulent forms. Major *R* genes, such as *Lr1*, *Lr10*, and *Lr21*, typically encode NLR (Nucleotide-binding, Leucine-rich repeat) proteins and trigger a rapid hypersensitive response [9]. In contrast, race non-specific resistance is known as adult-plant resistance (APR) or slow-rusting resistance. It involves multiple minor genes that work together [7,8]. APR reduces disease severity by prolonging the time before symptoms appear (latent period), lowering the number of infections, producing smaller spore masses (pustules), and reducing spore production [10]. This type of resistance works against all pathogen races, slowing down pathogen evolution and providing longer-lasting protection [3,10,11]. Examples include *Lr34* (encodes an ABC transporter) and *Lr67* (encodes a hexose transporter) with multiple effects and provide broad-spectrum, durable resistance against multiple rust diseases [12].

Significant progress has been made in understanding the genetic basis of leaf rust resistance in wheat. To date, over 100 leaf rust resistance genes (*Lr* genes) have been identified in bread wheat and its relatives, with 91 formally cataloged [13,14,15,16,17,18,19]. These genes are distributed on all 21 wheat chromosomes [20,21,22]. Most *Lr* genes provide ASR while others confer durable, race-non-specific APR. Examples of APR genes include the *Lr34/Yr18/Sr57* [23], *Lr46/Yr29/Sr58* [24], *Lr67/Yr46/Sr55* [25], *Lr68*, *Lr74*, *Lr75*, *Lr77*, and *Lr78* [6]. In China, ASR genes such as *Lr9*, *Lr19*, *Lr24*, *Lr28*, *Lr29*, *Lr38*, *Lr47*, *Lr51*, and *Lr53* remain effective [3]. However, their utilization is limited because they are sometimes linked to undesirable traits [3,5]. Genes like *Lr1*, *Lr2c*, *Lr3*, *Lr16*, *Lr17*, *Lr26*, *LrB*, *Lr3bg*, *Lr14b*, *Lr23*, and *Lr39* have largely lost effectiveness against current Pt races [3]. Cloning *Lr* genes is crucial to understand their function and enable precise breeding. Currently, at least 10 *Lr* genes have been cloned: including the *Lr1* (NLR), *Lr9/Lr58* (kinase/VWA domains), *Lr10* (NLR), *Lr13*, *Lr14a* (membrane protein), *Lr21* (NLR), *Lr22a* (NLR), *Lr42* (NLR), along with the pleiotropic APR genes *Lr34* and *Lr67*. Additionally, over 200 quantitative trait loci (QTL) for leaf rust resistance have been mapped across various wheat populations, which further increased the pool of potential resistance sources [3].

Given the growing threat of leaf rust under climate change and limitations of existing resistance genes, the need for more durable solutions, continuous discovery and employing new resistance sources is an urgent global priority. Therefore, identifying and cloning novel genes, uncovering their genetic mechanisms, and developing molecular tools for marker-assisted selection (MAS) breeding are crucial challenges. In this study, we employed a bi-parental population of 212 recombinant inbred lines (RILs) derived from a cross between Doumai (a landrace with moderated APR to leaf rust) and Shi 4185 (a high-yield variety with highly leaf rust APR). The objectives of this study are to: (1) identify novel resistance loci and tightly linked SNP markers through genetic analysis; and (2) develop robust, breeder-friendly kompetitive allele specific PCR (KASP) markers for wheat efficient MAS breeding for leaf rust resistance.

## 2. Results

### 2.1. Phenotypic Analysis

Statistical analysis of leaf rust maximum disease severity (MDS) was conducted for Doumai, Shi4185, and the RIL population at adult plant stages. Seedling inoculation results revealed that Doumai and Shi4185 exhibited moderate to high susceptibility (IT = 3–4) to the leaf rust races *THTT*, *PHTT*, and *THTS*. Adult plant stage evaluations showed that Shi4185 displayed higher resistance (MDS 18–32% across all 4 environments) to the leaf rust races, while Doumai exhibited moderate leaf rust resistance (MDS ranged from 22% to 52%). MDS of 212 lines in the RIL population across four environments ranged from 0% to 98%. The correlation coefficients of leaf rust resistance in the Doumai/Shi4185 RIL population across four environments ranged from 0.50 to 0.61 with significant correlations (*p* < 0.01). Under multiple environmental conditions, MDS showed a continuous distribution, indicating that the leaf rust resistance in Doumai and Shi4185 is controlled by multiple minor-effect genes. ANOVA indicated that the leaf rust resistance of Doumai/Shi4185 RIL population was significantly influenced by genotype, environment, and genotype-by-environment interactions (*p* < 0.01) with the leaf rust resistance broad-sense heritability (*H*_b_^2^) was 0.63, indicating a high level of genetic stability (Table 1).

### 2.2. QTL for Leaf Rust Resistance

We have identified five QTL associated with leaf rust resistance in Doumai / Shi4185 RIL population across four environments. Two QTL, *QLr.lfnu-1BL1* and *QLr.lfnu-1BL2*, on chromosome 1B were identified, and they were both detected in Xinxiang2021 and Zhengzhou2022. *QLr.lfnu-1BL1* was linked to *tplb0033h08_101* (484.2 Mb) and *D_contig03023_692* (487.6 Mb), and explained 6.24–7.15% of the phenotypic variance (PVE). *QLr.lfnu-1BL2* was linked to *Tdurum_contig84791_198* (674.0 Mb) and *Ex_c1058_1537* (678.3 Mb), explaining 6.85–8.67% of the PVE. On chromosome 2A, *QLr.lfnu-2AL* (772.9–779.9 Mb) identified in Zhengzhou2021 and Zhengzhou2022, flanked by *wsnp_JD_c289_450995* and *Excalibur_c40335_198*, and explained 6.75–6.98% of the PVE. Two QTLs were identified on chromosome 7B. *QLr.lfnu-7BL1* (675.7–681.8 Mb) effective in Xinxiang2021 and Zhengzhou2021, was close to *BobWhite_c7208_88* and *IACX8294*, with the PVE of 4.54–8.91%. *QLr.lfnu-7BL2* (701.3–708.0 Mb) detected in Xinxiang2022 and Zhengzhou2022, was linked to *tplb0058p02_2806* and *BobWhite_rep_c53128_119*, and could explain 4.97–7.41% of the PVE. In addition, the resistance alleles of *QLr.lfnu-1BL1* and *QLr.lfnu-1BL2* originated from Doumai, while the resistance alleles of *QLr.lfnu-2AL*, *QLr.lfnu-7BL1* and *QLr.lfnu-7BL2* were derived from Shi4185 (Table 2; Figure 1).

### 2.3. Additive Effect of QTL for Leaf Rust Resistance

To further investigate the comprehensive genetic effects of different leaf rust resistance QTL, the number of favorable alleles in the Doumai/Shi4185 RIL population across four environments was counted. The RILs were grouped based on the number of resistant alleles they carried (from one to four). Notably, no lines in the population carried all five resistant alleles or lacked any resistant alleles. The 212 lines were primarily divided into four groups, carrying 1, 2, 3, and 4 favorable alleles, respectively. Overall, with the number of resistance alleles increased, the MDS gradually decreased. Specifically, lines with one, two, three, and four resistance alleles had mean MDS of 49.0% (standard errors: 17.2 with 95% confidence intervals: 42.3–54.2%), 50.7% (standard errors: 14.9 with 95% confidence intervals: 43.1–56.0%), 45.2% (standard errors: 15.3 with 95% confidence intervals: 38.6–49.9%), and 31.5% (standard errors: 12.9 with 95% confidence intervals: 26.9–40.3%), respectively (Figure 2). This indicate that the five QTL exhibit additive effects, and wheat leaf rust resistance at adult plant stage can be strengthened by pyramiding multiple favorable alleles in breeding practice.

### 2.4. Validation of KASP Markers

To effectively utilize the QTL with stable and consistent effects on leaf rust resistance via MAS, we endeavored to transform the tightly linked SNP markers into *KASP-LR-1BL1* (*QLr.lfnu-1BL1*); *KASP-LR-2AL* (*QLr.lfnu-2AL*); *KASP-LR-7BL1* (*QLr.lfnu-7BL1*) and *KASP-LR-7BL2* (*QLr.lfnu-7BL2*). Among these, *KASP-LR-2AL* and *KASP-LR-7BL2* could distinguish two genotypes in both the Doumai/Shi4185 RIL population and natural population (Appendix A), while *KASP-LR-1BL1* and *KASP-LR-7BL1* could not be validated by natural population. In addition, we utilized *KASP-LR-2AL* and *KASP-LR-7BL2* to classify the 212 lines of Doumai/Shi4185 into four types, that is 2AS7BS, 2AR7BS, 2AS7BR and 2AR7BR. The genotypes carrying resistance alleles (2AR7BR) at both *QLr.lfnu-2AL* and *QLr.lfnu-7BL2* exhibited significantly lower MDS across all environments, and their mean MDS was also lower than those 2AR7BS, 2AS7BR, and 2AS7BS groups. In contrast, genotypes with both susceptibility alleles (2AS7BS) showed significantly higher MDS compared to the 2AR7BR, 2AS7BR, and 2AR7BS groups. Thus, *KASP-LR-2AL* and *KASP-LR-7BL2* can be effectively used in detecting resistant accessions in Doumai/Shi4185 RIL population. Thus, combining the resistance alleles of *QLr.lfnu-2AL* and *QLr.lfnu-7BL2* can significantly enhance resistance to leaf rust (Figure 3).

The natural population was genotyped using the *KASP-LR-2AL* and *KASP-LR-7BL2*. For *KASP-LR-2AL*, accessions possessing resistance alleles at *QLr.lfnu-2AL* exhibited significantly greater leaf rust resistance (99 lines with an average MDS of 47.9, standard errors: 18.7 with 95% confidence intervals: 44.2–51.6) compared to 43 lines with susceptible alleles (average MDS of 54.1, standard errors: 12.8 with 95% confidence intervals: 50.2–58.0%) (*p* < 0.05). Likewise, marked difference in MDS were observed between lines harboring the resistance allele (82 lines with an average MDS of 47.6, standard errors: 19.2 with 95% confidence intervals: 42.4–51.0%) and those with the susceptibility allele (68 lines with an average MDS of 53.1, standard errors: 13.9 with 95% confidence intervals: 50.3–57.1%) by *KASP-LR-7BL2*. Furthermore, the resistance alleles at *QLr.lfnu-2AL* and *QLr.lfnu-7BL2* were widespread, showing high frequencies of 66.0% and 54.6%, respectively, in the natural population (Table 3; Figure 4). 

### 2.5. Prediction of Candidate Genes for Leaf Rust Resistance

To identify potential candidate genes underlying the identified QTL, we analyzed the genomic intervals of the five stable QTLs using the Chinese Spring reference genome (IWGSC RefSeq v2.1). A total of 53 high-confidence genes were located within these regions. Based on functional annotation, these genes encode proteins known to be involved in plant disease resistance, including NLR proteins, receptor-like kinases, signaling kinases (e.g., CDPK), transcription factors (e.g., WRKY, NAC, MYB, bZIP), pathogenesis-related proteins (e.g., β-1,3-glucanase), and ABC transporters. We further refined the candidate gene list by examining their expression patterns in public databases (http://www.wheat-expression.com/, accessed on 18 July 2025), prioritizing those with high expression in leaves. This filtering step yielded a final set of 34 candidate genes with higher confidence. The distribution of these potential candidate genes across the QTLs is as follows: *QLr.lfnu-1BL1* (6 genes), *QLr.lfnu-1BL2* (10 genes), *QLr.lfnu-2AL* (6 genes), *QLr.lfnu-7BL1* (3 genes), and *QLr.lfnu-7BL2* (9 genes) (Figure 5; Appendix A).

## 3. Discussion

Since the 1970s, the International Maize and Wheat Improvement Center (CIMMYT) has made significant progress in research on APR to wheat rusts, and have successfully developed wheat varieties with durable resistance to rust diseases. The resistance of these varieties has remained effective over 50 years [2]. Recent advances in genomics and molecular breeding have provided powerful tools to dissect rust resistance [26]. By integrating QTL mapping, meta-analysis, and transcriptomic studies, researchers can identify stable and effective loci and candidate genes associated with durable resistance [27]. Excavating resistance genes from important/superior germplasms, developing molecular markers, and creating new germplasms are of great significance to promote wheat breeding for leaf rust resistance.

In this study, most of the QTLs identified were detected in two environments, and no QTLs were found to be present across all environments. The inconsistent detection of QTLs across environments is a common phenomenon in QTL mapping, often due to the interaction between genetic loci and the environment (G × E interaction). The variation in detection could be attributed to several factors, including: (1) For quantitative genetic traits in wheat (such as rust resistance, etc.), it is generally believed that QTLs detected in more than two environments exhibit a certain level of stability and can be utilized in wheat genetic breeding. Therefore, we typically select and report QTL that are present in two or more environments for further research and application. (2) Differences in disease pressure and environmental conditions (e.g., temperature, humidity) across the four trial environments, which can influence the expression of APR genes. (3) The possibility that some QTLs have minor effects that are statistically detectable only under specific environmental stresses. (4) The loci consistently detected in multiple environments are likely to be more stable and thus of higher priority for breeding applications, while those detected in only one or two environments may be conditionally effective.

### 3.1. Comparison with Previously Reported QTL

#### 3.1.1. *QLr.lfnu-1BL1* and *QLr.lfnu-1BL2*

Six known leaf rust resistance genes, *Lr26* [28], *Lr33* [29], *Lr44* [30], *Lr46* [30], *Lr51* [31], and *Lr75* [32], and five stable QTL, *QLr.sfr-1BS* [33], *QLr.pser-1BL* [34], *QLr.pbi-1B* [13], *QLr.caas-1BL*, and *QLr.hebau-1BL* [4], were detected on chromosome 1B. Among these genes, *Lr26*, *Lr33*, *Lr44*, and *Lr51* confer ASR, while *Lr46* and *Lr75* confer APR. *Lr55* distributed on chromosome 1BS was identified using SSR and DArT markers [35]. Dyck and Samborski [29] reported that *Lr33* provides resistance at both seedling and adult stages. Herrera-Foessel et al. [36] revealed that *Lr44* shows partial dominance to *Lr33*. Kuraparthy et al. [37] reported that *Lr71* was mapped near the centromere of chromosome 1B. Additionally, *QLr.sfr-1BS* originate from Forno was later designated as *Lr75* [32]. *QLr.pser-1BL* controls leaf rust resistance in Ning7840 [34]. *LrZH84* is another leaf rust resistance gene distributed on chromosome 1BL from Zhou8425B. *Lr46* (also known as *Yr29/Sr58/Pm39*) has been widely used in breeding, and several linked markers have been developed [30]. In addition, QTL detected in Bainong 64 were predicted to be *Lr46* [38,39]. *Lr51*, derived from *Ae. speltoides*, was mapped to chromosome 1BL [31]. *Lr26*, located on the 1BL/1RS translocation, may benefit from new rye genome resources. The QTL *QLr.pbi-1B* in Beaver was associated with the 1BL/1RS translocation [13]. Furthermore, Amo et al. [18] performed a meta-analysis on 320 leaf rust resistance QTL and identified a meta-locus, *MQTL1B.5* (flanked by *BS00000010_51* and *1103838*), on chromosome 1B at position 674.0–678.6 Mb. *MQTL1B.5* overlaps with *QLr.lfnu-1BL2* (flanked by *Tdurum_contig84791_198* and *Ex_c1058_1537*) at position 674.0–678.3 Mb identified in this study. However, no overlap was observed between *QLr.lfnu-1BL1* (flanked by *tplb0033h08_101-D_contig03023_692*, position 484.2–487.6 Mb) and the loci mentioned above. In addition, we compared our results with recently published studies on the genetics of leaf rust resistance in wheat. Based on reference genome alignment, no previously reported loci colocalizing with or closely linked to *QLr.lfnu-1BL1* were identified. [40,41,42,43]. Therefore, *QLr.lfnu-1BL1* may be novel.

#### 3.1.2. *QLr.lfnu-2AL*

Multiple leaf rust resistance genes and QTL distributed on chromosome 2A, including the *Lr17a* [44], *Lr37* [45], *Lr45* [46], *Lr65* [15], *Lr81* [16], and *LrM* [19], as well as the QTL *QLr.ifa-2AL* [47], *QLr.hebau-2AL* [48], and *QLr.spa-2A* [49]. Among these, *Lr17a*, *Lr37*, *Lr65*, *Lr81*, and *LrM* are located on the 2AS, while *Lr45* and *QLr.ifa-2AL* are mapped to 2AL. *Lr65* was fine-mapped to a 60.11 Kb region on 2AS [15]. *Lr37*, introgressed from *Aegilops ventricosa*, was initially mapped using the traditional RFLP and CAPS markers [45]. *Lr17a*, widely deployed in North American wheat varieties, has been overcome by emerging Pt pathotypes [44]. *Lr81* was localized to a 100 Kb region on 2AS [16]. *LrM*, derived from *Ae. markgrafii*, provides broad-spectrum resistance [19]. *Lr45*, introgressed from *Secale cereale*, was assigned to 2AS [46]. A GWAS analysis of 496 durum wheat accessions excavated *Lr-2AL* [50]. APR QTL *QLr.ifa-2AL*, *QLr.ifa-2BL*, and *QLr.ifa-3BS* were detected in the Australian cultivar Capo [47]. In Zhou 8425B, *QLr.hebau-2AL* and *QLr.hebau-4AL* were identified [48], whereas the Canadian cultivars AC Cadillac and Carberry contributed *QLr.spa-2A*, *QLr.spa-2B* and *QLr.spa-4B* [49]. Additionally, Amo et al. [18] performed a meta-QTL analysis and identified the meta-locus *MQTL2A.3* on 2A (729.9–752.3 Mb), which differs from *QLr.lfnu-2AL* (772.9–779.9 Mb) detected in this study. Additionally, we compared our findings with recently published genetic studies on leaf rust resistance in wheat. Based on comparisons using the reference genome, no genetic loci identical or closely linked to *QLr.lfnu-2AL* were identified [41,42,50,51]. Therefore, *QLr.lfnu-2AL* represents a novel locus for leaf rust resistance on 2AL.

#### 3.1.3. *QLr.lfnu-7BL1* and *QLr.lfnu-7BL2*

As reported, over 10 loci and genes for leaf rust resistance distributed on chromosome 7B, including *Lr14a* [17], *Lr68* [14], *Lr72* [52], and *LrFun* [53], *QLr.sfrs-7B.2* [39], *QLr.cimmyt-7BL1* [54], *QLr.osu-7BL* [55], *QLr.ubo-7B.2* [46], and *QLr.ksu-7BL* [56]. These loci are distributed across 7BS (*Lr72*) [47] and 7BL (*Lr14a*, *Lr68*, *LrFun*). *Lr14a* encodes a membrane-localized protein with ankyrin repeats and Ca^2+^-permeable ion channel domains [17]. *Lr68*, a slow-rusting gene from Parula, was mapped to the distal 7BL region [14]. *QLr.osu-7BL* showed consistent detection in CI13227 [55]. *QLr.ubo-7B.2* conferred APR for leaf rust in durum wheat Colosseo [51]. *LrFun* from Fundulea900 resides in the 7BL distal region [53]. Amo et al. [18] performed meta-QTL analysis and identified two meta loci on chromosome 7B, e.g., *MQTL7B.1* and *MQTL7B.2*. Among these, *QLr.lfnu-7BL1* (675.7–681.8 Mb), was overlapped with *MQTL7B.1* (675.1–679.3 Mb). In addition, the *MQTL7B.2* (*1114524-BobWhite_c2892_167*) (700.7–721.2 Mb) overlapped with *QLr.lfnu-7BL2* (701.3–708.0 Mb) identified in this study.

### 3.2. Prediction of Potential Candidate Genes for Leaf Rust Resistance

Wheat leaf rust has a complex genetic resistance mechanism. The cloned leaf rust resistance genes primarily encode key components of the plant immune system, whose core function is to recognize pathogens and activate defense responses. The core mechanisms include four categories: NLR proteins, acting as intracellular receptors to recognize effector proteins of *P. triticina* and form resistance bodies, initiating hypersensitive response that leads to cell death at infection sites, thereby limiting pathogen spread (e.g., *Lr1*, *Lr10*, *Lr21*) [3,20]. Membrane receptor proteins (RLKs/RLPs) perceiving pathogen-associated molecular patterns (PAMPs) or extracellular effector proteins, activating early defenses such as reactive oxygen species (ROS) burst and callose deposition through phosphorylation cascades. Signaling kinases (e.g., MAPK, CDPK) amplify and transmit defense signals, while transcription factors (WRKY, NAC, MYB, bZIP) coordinate the expression of defense-related genes [6,23,25]. Defense execution proteins (PR proteins, phytoalexin synthases), such as chitinases and antimicrobial proteins, directly inhibit pathogens. Additionally, candidate genes for resistance loci involve three functional categories: oxidative redox pathway genes (e.g., POD, PPO, LOX), which participate in phenolic metabolism, ROS regulation, and the synthesis of antimicrobial compounds; hormone signaling genes (e.g., F-box, serine/threonine kinases), which regulate defense hormone pathways such as SA, JA, and ET; and defense regulatory genes [3,6,20,21,23,25].

This study identified 53 potential candidate genes across five leaf rust resistance loci. These genes encode proteins including: NLR proteins, receptor-like kinases, signaling kinases (e.g., CDPK), transcription factors (e.g., WRKY, NAC, MYB, bZIP), pathogenesis-related proteins (e.g., β-1,3-glucanase), and ABC transporters. Using expression data from public databases, we further filtered and identified 34 potential candidate genes. The number of potential candidate genes corresponding to each resistance locus is as follows: *QLr.lfnu-1BL1* (6 potential candidate genes), *QLr.lfnu-1BL2* (10 potential candidate genes), *QLr.lfnu-2AL* (6 potential candidate genes), *QLr.lfnu-7BL1* (3 potential candidate genes), and *QLr.lfnu-7BL2* (9 potential candidate genes) (Figure 5; Appendix A).

Map-based cloning in wheat is hindered by its large (16 Gb), repetitive (>85%) allohexaploid (AABBDD) genome [27]. The existence of three homologous sub-genomes generates many homologous sequences near the target gene, which increases the difficulty of specific marker development and positional cloning [27]. The low recombination frequency also necessitates extremely large segregating populations (thousands to tens of thousands of individuals) for mapping. In this study, the identified loci spanned 3.4–7.0 Mb, which is too broad for direct gene identification. The list of resistance genes identified here provides a foundation for future research. It is important to note that these are only potential candidates, selected through computational analyses like annotation and expression profiling. While this approach is effective for generating hypotheses, it cannot prove the genes’ actual functions or roles in disease resistance. Future work on these loci will focus on: (1) constructing derivative populations from residual heterozygous lines; (2) developing KASP markers for fine mapping; (3) narrowing regions to <0.5 Mb and preliminarily identifying targets using annotation, variation, and expression data; and (4) functional validation via transgenic and gene editing techniques [19,57,58]. Final confirmation of their role requires direct experimental validation through genetic or transgenic studies. This distinction between prediction and confirmation is essential for rigorous scientific reporting.

### 3.3. Application in Wheat Breeding

Traditional breeding has enhanced wheat resistance to leaf rust. However, identifying this resistance in the field involves pathogen inoculation and maintaining specific disease conditions, which is challenging, costly, and time-consuming. This process leads to low efficiency in breeding efforts [59,60]. In recent years, advances in wheat genomics and molecular genetics have significantly accelerated wheat molecular breeding progress [27,61]. Particularly, the emergence of KASP markers has provided a high-throughput, cost-effective, and flexible tool for MAS breeding. In this study, five stable QTL were identified, and their aggregation demonstrated a favorable additive effect on leaf rust resistance. In the current study, we have successfully developed and validated two KASP markers, *KASP-LR-2AL* (*QLr.lfnu-2AL*) and *KASP-LR-7BL2* (*QLr.lfnu-7BL2*), which could serve as a molecular tool for wheat MAS breeding programs. However, it is also important to note that, due to the limitations imposed by the diversity of genetic backgrounds and environmental influences, these two KASP markers may also have limitations in wheat breeding. Additionally, several varieties with resistance alleles at *QLr.lfnu-2AL* and *QLr.lfnu-7BL2* loci perform excellent leaf rust resistance and agronomic traits, such as Huaimai21, Linkang 12, Shan512, Shannong78-59, Taishan5, Xinmai19, Xinmai9408, Qinnong731, Wanmai29, Xiaoyan54, Zhoumai18, Gaoyou503 and Jimai20, can also be used as excellent parental lines for wheat breeding.

## 4. Methods

### 4.1. Plant Materials and Phenotypic Evaluation

Doumai exhibits moderate susceptibility to moderate resistance against leaf rust at the adult plant stage, while Shi4185 shows moderate to high resistance. The Doumai/Shi4185 RIL population was cultivated in Xinxiang, Henan (34°53′ N, 113°23′ E) and Zhengzhou, Henan (34°74′ N, 113°62′ E) during the 2020–2021 and 2021–2022 growing seasons. The experimental design consisted of 2 m long rows with a randomized complete block design and three replicates. Zhengzhou5389 was used as the susceptible control and spreader row. Susceptible controls were planted every 10 rows in the field. A natural population comprising 150 accessions was employed to validate the genetic effects of leaf rust resistance QTL. This natural population was planted in Baoding of Hebei (38°86’ N, 115°48′ E) and Zhengzhou of Henan (34°74′ N, 113°62′ E) during the 2020–2021 and 2021–2022 growing seasons. The experiment was a completely randomized block design with three replicates, 2 m long rows, and 20 cm row spacing.

To ensure consistent pathogen pressure across different environments, inoculation was performed using a mixture of leaf rust spores. Xinxiang and Zhengzhou in Henan are regions with high incidence of wheat leaf rust, where the prevalent physiological races include *THTT*, *PHTT*, and *THTS*. At the jointing stage of wheat, urediniospores of *THTT*, *PHTT*, and *THTS* were mixed in equal proportions, and a few drops of 0.03% Tween 20 solution were added before spraying. We assessed disease severity (DS), defined as the percentage of leaf area covered by leaf rust urediniospore pustules, using a single-row plot design with 30 plants per row to evaluate the overall disease response of the group. Approximately four weeks after inoculation, when the DS of the control variety Zhengzhou5389 approached 100% with nearly the entire leaf surface covered, disease evaluations were initiated. Field assessments were conducted weekly for the population and its parents, with the process repeated 2–3 times. The MDS observed across these assessments in each environment was selected as the phenotypic data for subsequent linkage analysis. Biological replicates were averaged prior to analysis.

### 4.2. Statistical Analyses and Linkage Mapping

Analysis of variance (ANOVA) was performed using IciMapping v4.2 (http://www.isbreeding.net/, accessed on 18 July 2025) [62]. The Pearson correlation coefficient was used to evaluate the correlation of MDS. The broad-sense heritability (*H*_b_^2^) of leaf rust resistance was calculated using the formula: *H*_b_^2^ = *σ*_g_^2^/(*σ*_g_^2^ + *σ*_ge_^2^/e + *σ*_ε_^2^/re), where *σ*_g_^2^ represents genotypic variance, *σ*_ge_^2^ represents genotype-by-environment interaction variance, and *σ*_ε_^2^ represents error variance. Here, e denotes the number of environments, and r denotes the number of replicates.

The wheat 90K SNP chip was used to genotype Doumai, Shi4185, and all lines within the RIL population. A total of 80,587 SNPs were detected, and the genetic map was constructed by Wen et al. [63]. This map contains 9354 high-quality polymorphic markers. Inclusive composite interval mapping (ICIM) in IciMapping v2.1 was employed for QTL mapping of MDS in the Doumai/Shi4185 RIL population across four environments. This method combines interval mapping and multiple regression analysis. A significance level (*p* = 0.05) was determined through 1000 permutation tests, with an LOD threshold set at 2.5 and a step size of 1.0 cM. LOD value curves and QTL position information was generated. In this study, QTL detected in two or more environments were defined as stable QTL.

### 4.3. Development of KASP Markers

KASP primers were developed for the flanking and intra-interval markers of the identified leaf rust resistance QTL. The primers were designed using the PolyMarker website (http://www.polymarker.info/, accessed on 18 July 2025) (Table A1). KASP primers consist of two competitive primers and one common primer, where the two competitive primers differ by only one base at the 3’ end. Specific or semi-specific KASP primers for the locus were selected, and FAM and HEX tag sequences were added to the 5′ ends of the competitive primers, respectively. The primer sequences were submitted to Sangon Biotech (Shanghai, China) Co., Ltd. (https://store.sangon.com/, accessed on 18 July 2025) for synthesis. The KASP marker PCR amplification system was as follows: 2.0 µL KASP 2× Master Mix, 0.0336 µL KASP primers, and 2 µL template DNA (50 ng/µL). The amplification reaction was performed using a water bath PCR instrument from LGC (Laboratory of the Government Chemist, London, England). The specific reaction program was as follows: initial denaturation at 94 °C for 15 min (1 cycle), followed by 10 cycles of denaturation at 94 °C for 20 s and annealing at a temperature gradient from 65 °C to 57 °C for 1 min. Finally, 35 cycles of denaturation at 94 °C for 20 s and annealing at 57 °C for 1 min were performed. The temperature and time were gradually adjusted to ensure effective DNA amplification. After the amplification program, the sample plate was scanned using the 384 KASP program on the PHEstar instrument (BMG Labtech GmbH, Ortenberg, Germany) to read fluorescence values. The genotyping results were visualized using KLUSTER V1.0 (LGC, Hoddesdon, UK). Typically, if FAM fluorescence alone was detected, the amplification product emitted blue fluorescence; if HEX fluorescence alone was detected, the product emitted red fluorescence; and if FAM and HEX fluorescence were both detected, the product emitted green fluorescence.

### 4.4. Candidate Gene Prediction

Candidate genes were selected based on gene annotation information from the Chinese Spring reference genome (IWGSC RefSeq v2.1). The wheat gene expression database (http://www.wheat-expression.com/, accessed on 18 July 2025) was used to analyze the expression patterns of candidate genes and identify those specifically and highly expressed in leaves. The specific steps were as follows: First, high-confidence candidate genes within the identified QTL regions were screened using the annotation information provided by IWGSC RefSeq v2.1. Second, the candidate gene list was further simplified by including proteins encoded by reported cloned genes related to wheat fungal diseases, such as lectin receptor kinase proteins and ABC transporter proteins. Third, existing public databases were used to determine the expression patterns and expression tissues of the candidate genes. The above three criteria provide a preliminary candidate gene.

## 5. Conclusions

This study identified five APR QTLs for leaf rust resistance in wheat, each explaining 4.54–8.91% of the PVE. *QLr.lfnu-1BL1* and *QLr.lfnu-2AL* are likely novel loci, enriching the genetic resources available for breeding. We developed two breeder-friendly KASP markers, validated in a natural population, demonstrating their potential for MAS breeding. Furthermore, 34 genes within the QTL intervals were prioritized as candidate genes. While the individual QTL effects are moderate, their pyramiding could enhance resistance durability. Future work should focus on fine-mapping the novel QTLs, validating these markers in diverse genetic backgrounds, and functionally characterizing the leading candidate genes. This study provides foundational resources for developing wheat cultivars with durable leaf rust resistance.

## Figures and Tables

**Figure 1 plants-14-03113-f001:**
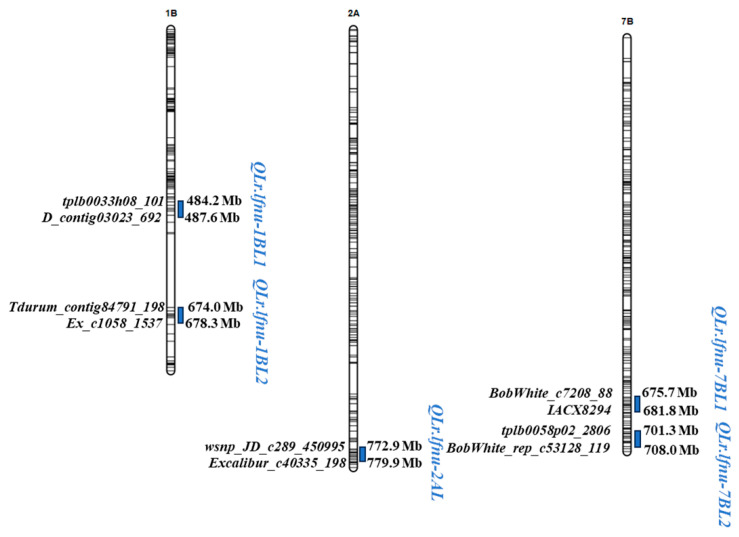
The QTL for leaf rust resistance identified in Doumai/Shi4185 RIL population.

**Figure 2 plants-14-03113-f002:**
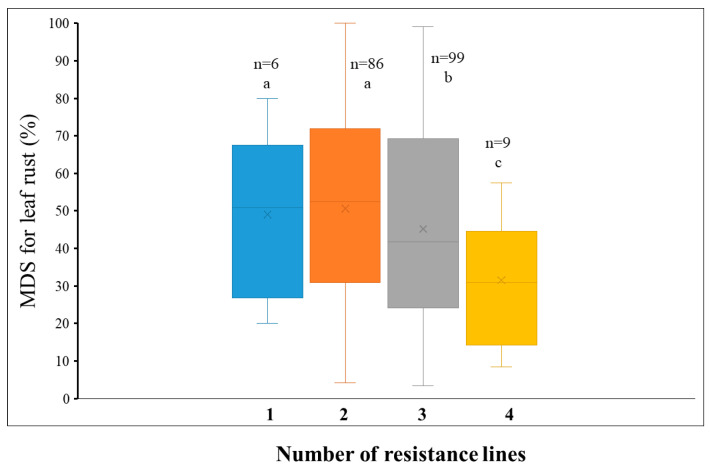
The effects of favorable allele combinations on maximum disease severity (MDS) of leaf rust among different classes in the Doumai/Shi4185 RIL population. The number of favorable alleles combined in each subset of RIL. Different alphabets indicate significant difference at *p* < 0.05.

**Figure 3 plants-14-03113-f003:**
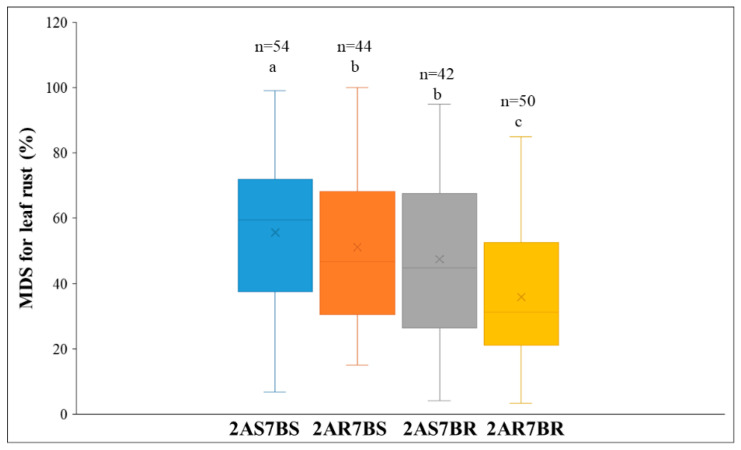
Validation of KASP markers *QLr.lfnu-2AL* and *QLr.lfnu-7BL* in the Doumai/Shi4185 RIL population. n means the number of lines. Different alphabets indicate significant difference at *p* < 0.05. MDS: means the maximum disease severity; 2AS or 2AR means the combination of the resistance or susperiatble allelel of *QLr.lfnu-2AL;* 7BS or 7BR means the combination of the resistance or susperiatble allelel of *QLr.lfnu-7BL*.

**Figure 4 plants-14-03113-f004:**
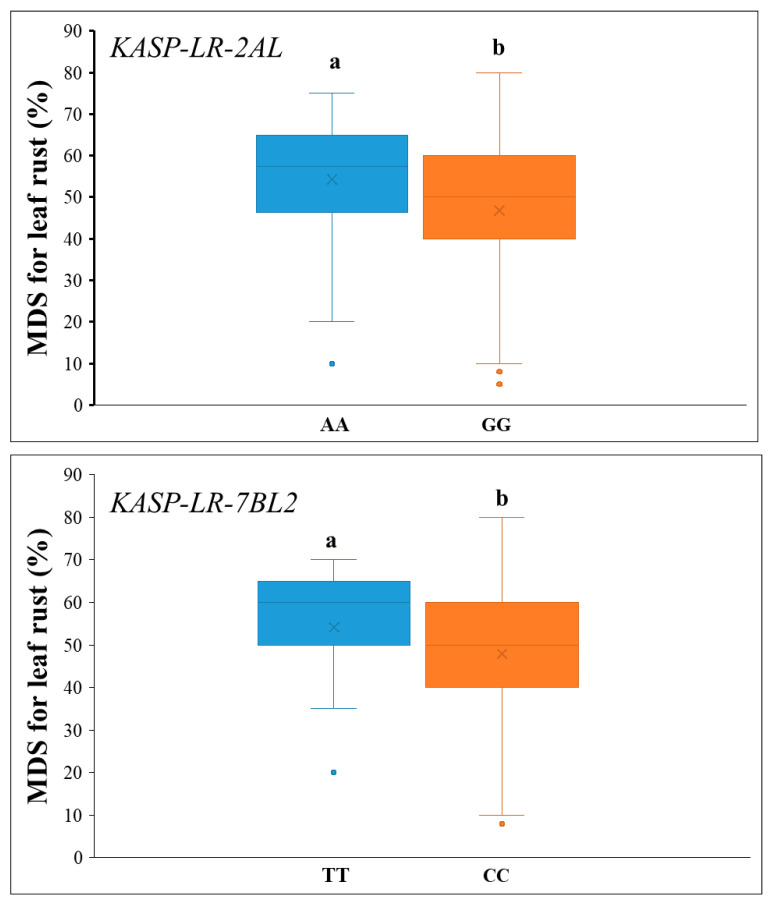
Validation of QTL *QLr.lfnu-2AL* and *QLr.lfnu-7BL2* in the panel of 150 wheat cultivars from the Huang-Huai River Valleys region. Different alphabets indicate significance at *p* < 0.05. MDS: maximum disease severity for leaf rust.

**Figure 5 plants-14-03113-f005:**
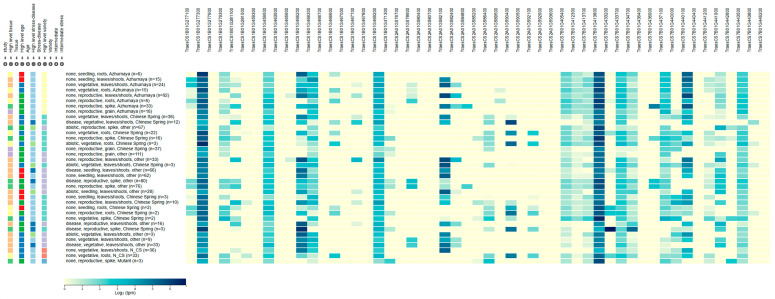
Expression pattern of the candidate genes for leaf rust resistance (https://www.wheat-expression.com/).

**Table 1 plants-14-03113-t001:** Summary of the MDS for leaf rust resistance in Doumai/Shi4185 RIL population.

Environment	Doumai	Shi4185	Mean MDS (%)	Range of MDS (%)	Standard Deviation	Coefficient of Variation
Xinxiang2021	48.0	38.0	43.3	0–98	33.2	0.8
Xinxiang2022	52.0	32.0	53.8	0–95	28.8	0.5
Zhengzhou2021	50.0	35.0	52.2	0–96	33.8	0.6
Zhengzhou2022	22.0	18.0	34.9	1–98	28.3	0.8

**Table 2 plants-14-03113-t002:** The QTL for leaf rust resistance identified in Doumai/Shi4185 RIL population.

QTL	Chromosome	Marker Interval	Physical Interval (Mb)	LOD	PVE (%) ^a^	Additive	Environment ^b^
*QLr.lfnu-1BL1*	1B	*tplb0033h08_101~* *D_contig03023_692*	484.2–487.6	2.92–3.56	6.24–7.15	−7.82–9.04	E1, E3
*QLr.lfnu-1BL2*	1B	*Tdurum_contig84791_198~* *Ex_c1058_1537*	674.0–678.3	3.38–3.98	6.85–8.67	−8.58–9.97	E1, E4
*QLr.lfnu-2AL*	2A	*wsnp_JD_c289_450995~* *Excalibur_c40335_198*	772.9–779.9	1.84–1.96	6.75–6.98	7.57–8.56	E3, E4
*QLr.lfnu-7BL1*	7B	*BobWhite_c7208_88~* *IACX8294*	675.7–681.8	2.71–3.54	4.54–8.91	6.75–8.57	E1, E3
*QLr.lfnu-7BL2*	7B	*tplb0058p02_2806~* *BobWhite_rep_c53128_119*	701.3–708.0	2.41–3.55	4.97–7.41	7.53–9.20	E2, E4

^a^ PVE: Phenotypic variance explained (%); ^b^ E1: Xinxiang at 2021–2022 cropping season; E2: Xinxiang at 2022–2023 cropping season; E3: Zhengzhou at 2021–2022 cropping season; E4: Zhengzhou at 2022–2023 cropping season.

**Table 3 plants-14-03113-t003:** *KASP-LR-2AL* and *KASP-LR-7BL2* validated in the diverse panel.

QTL	KASP Marker Name	Genotype	No. of Lines	Maximum Disease Severity (%)	*p*-Value
*QLr.lfnu-2AL*	*KASP-LR-2AL*	AA	43	54.1	0.049
		GG	99	47.9	
*QLr.lfnu-7BL2*	*KASP-LR-7BL2*	TT	68	53.1	0.007
		CC	82	47.6	

## Data Availability

Data are contained within the article and Appendix A.

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
