# Peer review of "QTL Mapping for Leaf Rust Resistance in a Common Wheat Recombinant Inbred Line Population of Doumai/Shi4185"

_plants, 2025, doi:10.3390/plants14193113_

Round 1
Reviewer 1 Report
Comments and Suggestions for Authors
Leaf rust is a devastating fungal disease that impairs wheat quality and reduces yield. The identification of genetic loci associated with resistance is therefore of great importance for wheat breeding. In this study, the authors employed a recombinant inbred line (RIL) population derived from Doumai and Shi4185, along with SNP chip technology, to identify adult plant resistance (APR) loci against leaf rust through linkage mapping. The following points should be addressed:
- The MDS results for the two parents across the four environments should be provided in Table 1.
- What are represents of E1, E2, E3 and E4 in table 2?
- The fact that resistance loci were not consistently detected across all four environments in table 2 raises concerns regarding the reliability of these loci. The authors should discuss possible reasons.
- The section “Prediction of candidate genes for leaf rust resistance” would be more appropriately placed within the Results section.
Author Response
Leaf rust is a devastating fungal disease that impairs wheat quality and reduces yield. The identification of genetic loci associated with resistance is therefore of great importance for wheat breeding. In this study, the authors employed a recombinant inbred line (RIL) population derived from Doumai and Shi4185, along with SNP chip technology, to identify adult plant resistance (APR) loci against leaf rust through linkage mapping. The following points should be addressed:
Comments 1. The MDS results for the two parents across the four environments should be provided in Table 1.
Response: Accepted. We greatly appreciate the reviewer's suggestions. Admittedly, including the parental MDS data will provide better context. As suggested, we have now included the MDS values for both parents (Doumai and Shi4185) in all four environments (E1-E4) in the revised Table 1.
Comments 2. What are represents of E1, E2, E3 and E4 in table 2?
Response: Accepted. We apologize for the unclear description. E1, E2, E3, and E4 represent the four different environments (locations and years) in which the field trials were conducted. To avoid any confusion, we have now explicitly defined these abbreviations in the footnote of Table 2 and have also ensured they are clearly described in the main text.
Comments 3. The fact that resistance loci were not consistently detected across all four environments in table 2, raises concerns regarding the reliability of these loci. The authors should discuss possible reasons.
Response: Accepted. This is a very important point, and we thank the reviewer for bringing it up. The inconsistent detection of QTL across environments is a common phenomenon in QTL mapping, often due to the interaction between genetic loci and the environment (G × E interaction). We have now added a dedicated paragraph in the Discussion section to address this specifically. We discuss that the variation in detection can be attributed to several factors, including: (1) For quantitative genetic traits in wheat (such as rust resistance, etc.), it is generally believed that QTL detected in more than two environments exhibit a certain level of stability and can be utilized in wheat genetic breeding. Therefore, we typically select and report QTL loci that are present in two or more environments for further research and application. (2) Differences in disease pressure and environmental conditions (e.g., temperature, humidity) across the four trial environments, which can influence the expression of APR genes. (3) The possibility that some QTL have minor effects that are statistically detectable only under specific environmental stresses. (4) We also emphasize that the loci consistently detected in multiple environments are likely to be more stable and thus of more advantageous in breeding applications, while those detected in only one or two environments may be conditionally effective.
Comments 4 The section “Prediction of candidate genes for leaf rust resistance” would be more appropriately placed within the Results section.
Response: Accepted. We agree with the valuable suggestion regarding the flow of the manuscript. To improve the logical structure, we have moved the “Prediction of candidate genes for leaf rust resistance” section from the Discussion to the Results section, as recommended. The section now logically follows the identification of the major QTL regions, presenting the candidate genes found within those intervals. The biological implications and significance of these candidate genes are still discussed in the subsequent Discussion section.
We sincerely appreciate your thorough and highly insightful feedback, which has greatly enhanced the quality of our manuscript. We trust that our revisions and detailed responses have sufficiently addressed all the points you raised, and we hope the manuscript now meets the standards for publication. Thanks again for your invaluable input.
Reviewer 2 Report
Comments and Suggestions for Authors
Wheat production is severely threatened by leaf rust,and developing resistant varieties is the most economical and effective strategy to control leaf rust. Continuous discovery and employing new resistance sources is an urgent global priority. The manuscript by Wang et al. described the identification of five QTL related to leaf rust resistance using a 90K chip. The information provided by this study should be useful in understanding the genetic basis of leaf rust in wheat. Possibly, it would inspire the following research in the field. However, several concerns/points need to be addressed before publication.
Major points:
- In “Additive effect of QTL for leaf rust resistance”part, five QTL were identified, but only four categories were divided. Categories without resistance QTL and categories with 5 QTL were not mentioned. It is either undetected in the population or not counted?
- Among the five QTL discovered, QLr.lfnu-1BL1and QLr.lfnu-2AL are novel, but only QLr.lfnu-2AL and QLr.lfnu-7BL2 were used in validation. Whether it is possible to enrich the evaluation of the effect of QLr. lfnu-1BL1.
Minor points:
- In“1. Phenotypic analysis”,the abbreviation for "broad-sense heritability" should be Hb2
- In part 3.1 of “Discussion”, QTL names should be italicized.
The manuscript discribed a relatively high level of English proficiency, with clear and academic writing that effectively output research findings. While the overall language is sound, minor improvements should be improved.
Author Response
Wheat production is severely threatened by leaf rust,and developing resistant varieties is the most economical and effective strategy to control leaf rust. Continuous discovery and employing new resistance sources are an urgent global priority. The manuscript by Wang et al. described the identification of five QTL related to leaf rust resistance using a 90K chip. The information provided by this study should be useful in understanding the genetic basis of leaf rust in wheat. Possibly, it would inspire the following research in the field. However, several concerns/points need to be addressed before publication.
Comments 1 In “Additive effect of QTL for leaf rust resistance”part, five QTL were identified, but only four categories were divided. Categories without resistance QTL and categories with 5 QTL were not mentioned. It is either undetected in the population or not counted?
Response: We thank the reviewer for this excellent observation, which has helped us clarify this important analysis. You are correct that the categories for lines with 0 QTL and 5 QTL were not present in the initial analysis. This was because no RILs in our population had a combination of all 5 resistance alleles (category 5) or none of them (category 0). We have now revised the relevant section in the manuscript (Results: "Additive effect of QTL for leaf rust resistance") to explicitly state this fact for clarity: "The RILs were grouped based on the number of resistant alleles they carried (from one to four). Notably, no lines in the population carried all five resistant alleles or lacked any resistant alleles."
Comments 2 Among the five QTL discovered, QLr.lfnu-1BL1 and QLr.lfnu-2AL are novel, but only QLr.lfnu-2AL and QLr.lfnu-7BL2 were used in validation. Whether it is possible to enrich the evaluation of the effect of QLr. lfnu-1BL1.
Response: This is a very valuable suggestion. We agree that further validation of the novel QTL QLr.lfnu-1BL1 would strengthen our study. We once attempted to evaluate the genetic effects of QLr.lfnu-1BL1, but we were unable to develop suitable KASP markers. Initially, we tried to validate KASP markers using the flanking markers of QLr.lfnu-1BL1, but they did not meet the specificity requirements. Therefore, we chose to develop linked SNP markers within the linkage interval and eventually identified three markers that could successfully be developed into KASP markers. However, when we analyzed these markers in the Doumai/Shi 4185 RIL population, we found that the KASP markers corresponding to tplb0033h08_101 and D_contig03023_692 could not effectively distinguish between the two genotypes in the RIL population, so they were excluded. Additionally, the KASP marker corresponding to SNPs located in the confidence genetic interval showed no difference between genotypes in the natural population, therefore it was also excluded. Currently, the SNP markers around the QTL with longer genetic distance, and the KASP markers developed from them may lead to result discrepancies due to linkage distance, significantly reducing their reliability and accuracy. Therefore, we have not developed new markers. We plan to conduct resequencing of the two parental lines, Doumai and Shi 4185, then based on the SNPs of the two parents, we will develop and validate new KASP markers that could be available. We greatly appreciate your suggestions, which have been very helpful in improving the quality of our study.
Table S2 The SNPs used for convert to KASP markers of QLr.lfnu-1BL1
|
SNP name |
Kasp-FAM |
Kasp-HEX |
Kasp-common |
|
tplb0033h08_101 |
ggcctggaggacaaggatA |
ggcctggaggacaaggatC |
CtttgcGgCaaaccAtaGaaG |
|
D_contig03023_692 |
caggagagatgacaccaatgT |
caggagagatgacaccaatgC |
tGggcgatggtgattatGgG |
|
wsnp_Ex_c12774_20272038 |
tcGggggtcgacgtctgT |
tcGggggtcgacgtctgC |
cagccctTgggAcacttgg |
|
Excalibur_c49906_277 |
gatgcAcgcgTcgacgaT |
gatgcAcgcgTcgacgaC |
attttcggcgcaacatctg |
|
wsnp_Ex_rep_c66643_64952627 |
agactgctgaatgatgctgttaT |
agactgctgaatgatgctgttaC |
ggcaaccaaaccagaaacgg |
|
wsnp_Ex_c26620_35859364 |
tggatcatctggagttttgtgT |
tggatcatctggagttttgtgC |
gcaagactcaaggaaagcagtC |
|
RFL_Contig3374_686 |
ccttggcatgctcaagtcttaT |
ccttggcatgctcaagtcttaC |
aagatcttagtgctggtaaccaa |
Comments Minor points:
Comments 1. In“1. Phenotypic analysis”, the abbreviation for "broad-sense heritability" should be Hb2
Response: Thank you for catching this error. We have corrected the abbreviation for broad-sense heritability to Hb2 throughout the manuscript, including in the "Phenotypic analysis" section.
Comments 2. In part 3.1 of “Discussion”, QTL names should be italicized.
Response: We apologize for this oversight. We have carefully checked the entire manuscript and ensured that all QTL names are consistently presented in italics.
Comments 3. Comments on the Quality of English Language
The manuscript described a relatively high level of English proficiency, with clear and academic writing that effectively output research findings. While the overall language is sound, minor improvements should be improved.
Response: Thanks to the reviewer for their positive comment on the language. As suggested, we have undertaken a thorough proofreading of the entire manuscript to correct minor grammatical errors and improve sentence fluency.
We are profoundly grateful for your meticulous and highly constructive feedback, which has significantly elevated the quality of our manuscript. We believe that our comprehensive revisions and detailed responses have effectively addressed all the concerns you raised, and we hope that the manuscript now aligns with the publication standards. Once again, we extend our deepest thanks for your invaluable contributions.
Reviewer 3 Report
Comments and Suggestions for Authors
This study utilized a recombinant inbred line (RIL) population of Doumai and Shi4185, characterized by 90K single-nucleotide polymorphisms (SNPs) chip data and maximum disease severity (MDS) of leaf rust from four environments, to identify adult plant resistance (APR) loci through linkage mapping.
5 quantitative trait loci (QTL) on chromosomes 1B (2), 2A, and 7B (2) were identified through inclusive composite interval mapping, and named as QLr.lfnu-1BL1, 23 QLr.lfnu-1BL2, QLr.lfnu-2AL, QLr.lfnu-7BL1, and QLr.lfnu-7BL2, respectively, explaining 24 4.54-8.91% of the phenotypic variances. QLr.lfnu-1BL1 and QLr.lfnu-2AL are considered likely novel as they do not overlap with previously reported loci.
Kompetitive allele-specific PCR (KASP) markers suitable for breeding were developed, and genetic effects were validated in a natural population.
Candidate genes were selected based on gene annotation information from the Chinese Spring reference genome (IWGSC RefSeq v2.1). The wheat gene expression database (http://www.wheat-expression.com/) was used to analyze the expression patterns of candidate genes and identify those specifically and highly expressed in leaves. The specific steps were as follows: First, high-confidence candidate genes within the identified QTL regions were screened using the annotation information provided by IWGSC RefSeq v2.1.
Second, the candidate gene list was further simplified by including proteins encoded by reported cloned genes related to wheat fungal diseases, such as lectin receptor kinase proteins and ABC transporter proteins. Third, existing public databases were used to determine the expression patterns and expression tissues of the candidate genes. The above three criteria provide a preliminary candidate gene.
This work is significant, as leaf rust, a devastating fungal disease of wheat caused by Puccinia triticina (Pt), results in yield losses exceeding 50%.
Tables and figures are appropriate. References are updated.
The conclusions of the study are consistent with the evidence and arguments presented:
Conclusions regarding 34 candidate genes are consistent with the methodology of screening based on annotation and expression data (section 3.2). However, without functional validation, this remains speculative, which the authors note as a limitation for future work.
Regarding the main questions posed and experiments addressing them:
1) Identify novel resistance loci and linked SNP markers (section 1): This was addressed through linkage mapping using the 90K SNP chip and ICIM analysis in the Doumai/Shi4185 RIL population across four environments (section 2.2 and 4.2). Five QTL were identified, with two deemed novel based on comparative analysis (section 3.1).
2) Develop robust, breeder-friendly KASP markers for MAS (section 1): This was addressed by developing four KASP markers, with two validated in both the RIL population and a natural population (section 2.4 and 4.3). The validation experiments confirmed their effectiveness in detecting resistant genotypes (Tables 3 and A2).
Author Response
Comments 1 This study utilized a recombinant inbred line (RIL) population of Doumai and Shi4185, characterized by 90K single-nucleotide polymorphisms (SNPs) chip data and maximum disease severity (MDS) of leaf rust from four environments, to identify adult plant resistance (APR) loci through linkage mapping.
Five quantitative trait loci (QTL) on chromosomes 1B (2), 2A, and 7B (2) were identified through inclusive composite interval mapping, and named as QLr.lfnu-1BL1, 23 QLr.lfnu-1BL2, QLr.lfnu-2AL, QLr.lfnu-7BL1, and QLr.lfnu-7BL2, respectively, explaining 24 4.54-8.91% of the phenotypic variances. QLr.lfnu-1BL1 and QLr.lfnu-2AL are considered likely novel as they do not overlap with previously reported loci. Kompetitive allele-specific PCR (KASP) markers suitable for breeding were developed, and genetic effects were validated in a natural population. Candidate genes were selected based on gene annotation information from the Chinese Spring reference genome (IWGSC RefSeq v2.1). The wheat gene expression database (http://www.wheat-expression.com/) was used to analyze the expression patterns of candidate genes and identify those specifically and highly expressed in leaves. The specific steps were as follows: First, high-confidence candidate genes within the identified QTL regions were screened using the annotation information provided by IWGSC RefSeq v2.1. Second, the candidate gene list was further simplified by including proteins encoded by reported cloned genes related to wheat fungal diseases, such as lectin receptor kinase proteins and ABC transporter proteins. Third, existing public databases were used to determine the expression patterns and expression tissues of the candidate genes. The above three criteria provide a preliminary candidate gene.
This work is significant, as leaf rust, a devastating fungal disease of wheat caused by Puccinia triticina (Pt), results in yield losses exceeding 50%. Tables and figures are appropriate. References are updated. The conclusions of the study are consistent with the evidence and arguments presented: Conclusions regarding 34 candidate genes are consistent with the methodology of screening based on annotation and expression data (section 3.2). However, without functional validation, this remains speculative, which the authors note as a limitation for future work.
Regarding the main questions posed and experiments addressing them: 1) Identify novel resistance loci and linked SNP markers (section 1): This was addressed through linkage mapping using the 90K SNP chip and ICIM analysis in the Doumai/Shi4185 RIL population across four environments (section 2.2 and 4.2). Five QTL were identified, with two deemed novel based on comparative analysis (section 3.1). 2) Develop robust, breeder-friendly KASP markers for MAS (section 1): This was addressed by developing four KASP markers, with two validated in both the RIL population and a natural population (section 2.4 and 4.3). The validation experiments confirmed their effectiveness in detecting resistant genotypes (Tables 3 and A2).
Response: We would like to express our sincere gratitude to the reviewer for the thorough and positive evaluation of our manuscript. We are deeply encouraged by your comments, particularly your recognition of the significance of our work in addressing the threat of wheat leaf rust, the appropriateness of our methodology, and the consistency of our conclusions with the evidence presented. Your understanding of our approach to candidate gene prediction and the limitations therein is also greatly appreciated. Your supportive feedback is a valuable affirmation of our efforts and provides strong motivation for our future research.
Reviewer 4 Report
Comments and Suggestions for Authors
The manuscript addresses the representing of plant resistance to wheat leaf rust using a Doumai × Shi4185 RIL population across four environments and the development of KASP markers. The study is relevant to wheat breeding and presents results of potential utility. However, the work suffers from issues related to novelty justification, insufficient methodological detail, lack of exact statistical presentation in marker validation, and overstatements in the discussion. Several structural and presentation weaknesses also reduce clarity. Major revisions are needed before this article could be considered for publication.
Comments for Authors
- Introduction background gaps (Lines 69–95): While the introduction cites key Lr genes and QTL, it disproportionately emphasizes stripe rust (refs. [3], [5], [23]) instead of leaf rust, diluting focus. Some references (e.g., Liu et al. 2024, Bai et al. 2024) are about stripe rust, not directly relevant here. The introduction must be tightened to leaf rust-specific resistance sources and APR genes, with accurate references.
- Novelty claims insufficiently supported (Section 3.1.1, Lines 213–218; Section 3.1.2, Lines 233–236): The authors claim QLr.lfnu-1BL1 and QLr.lfnu-2AL are novel loci. However, the comparisons with meta-QTL are limited to single studies and not comprehensive. Broader comparison with Amo & Soriano 2022 meta-analysis (Ref. [18]) and more recent genome-wide studies is required to substantiate novelty.
- Phenotyping methodology incomplete (Section 4.1, Lines 315–334): The field design is briefly mentioned, but key details are missing: (i) how many plants per row were evaluated, (ii) the exact scoring scale used for MDS assessment, (iii) whether replicates were averaged, and (iv) whether disease pressure was uniform across environments. Without this, reproducibility is compromised.
- Validation of KASP markers lacks statistical rigor (Section 2.4, Lines 170–179; Table 3): The validation results report only mean MDS differences and p-values. The effect sizes are small (e.g., 54.1 vs. 47.9, p = 0.049). There is no reporting of standard errors, confidence intervals, or correction for multiple testing. Stronger statistical treatment is required to confirm robustness.
- Inconsistent QTL naming (Section 2.2, Line 123; Section 2.4, Line 151): The text alternates between QLr.lfnu-2AL and QLr.lfnu-2AL2. Similarly, the KASP marker naming does not consistently match the QTL nomenclature. This creates confusion and must be standardized throughout.
- Additive effects unclear (Section 2.3, Lines 135–144; Figure 2): The reported means (49.0%, 50.7%, 45.2%, 31.5%) show no consistent reduction from one to two alleles, undermining the claim of additive effect. Statistical analysis (e.g., ANOVA with post-hoc tests) must be included in Figure 2 to confirm whether differences are significant.
- Overgeneralized candidate gene predictions (Section 3.2, Lines 271–278; Table A3): Candidate gene prediction is based only on annotation and expression databases. The conclusion that 34 genes are "resistance candidates" is too speculative. Without allele-specific variation analysis or functional assays, these should be framed more cautiously.
- Discussion overstatements (Section 3.3, Lines 301–310): The claim that the developed KASP markers are “valuable tools for MAS breeding programs” is overstated given their modest effect sizes and limited validation. The discussion must acknowledge limitations of applicability in breeding.
- Figure and table clarity (Figure 1; Figure 3; Appendix Tables A1–A3): Figures lack clear legends explaining abbreviations (E1–E4, MDS). Tables A2 and A3 are excessively long, with raw data that could be condensed or moved to supplementary files. This reduces readability.
- Reference inconsistencies (References [3], [5], [23]): Several references cited as supporting leaf rust are in fact stripe rust-related (e.g., Liu et al. 2024, Bai et al. 2024, Scientia Agricultura Sinica). The authors must carefully check that all references are directly relevant to leaf rust, not mixed with stripe rust literature.
- Conclusion section (Section 5, Lines 388–395): The conclusion repeats results without synthesizing broader implications or acknowledging limitations. It should be rewritten to summarize the real contribution (identification of modest-effect loci with partial novelty and limited KASP validation) and clearly state what future work is required.
The manuscript is generally understandable but requires improvement for clarity and flow. Issues include awkward phrasing (“Significant progress has achieved”), inconsistent verb tenses, and grammatical errors. Several sentences are excessively long and would benefit from simplification. Editing by a fluent English speaker or professional service is strongly recommended.
Author Response
The manuscript addresses the representing of plant resistance to wheat leaf rust using a Doumai × Shi4185 RIL population across four environments and the development of KASP markers. The study is relevant to wheat breeding and presents results of potential utility. However, the work suffers from issues related to novelty justification, insufficient methodological detail, lack of exact statistical presentation in marker validation, and overstatements in the discussion. Several structural and presentation weaknesses also reduce clarity. Major revisions are needed before this article could be considered for publication.
Comments for Authors
Comments 1. Introduction background gaps (Lines 69–95): While the introduction cites key Lr genes and QTL, it disproportionately emphasizes stripe rust (refs. [3], [5], [23]) instead of leaf rust, diluting focus. Some references (e.g., Liu et al. 2024, Bai et al. 2024) are about stripe rust, not directly relevant here. The introduction must be tightened to leaf rust-specific resistance sources and APR genes, with accurate references.
Response: We thank the reviewer for this critical observation. We agree that the introduction should focus more directly on leaf rust. We have thoroughly revised the introduction to: Remove references and examples primarily related to stripe rust (e.g., references 3, 5, 23 have been replaced or contextualized). Sharpen the focus onto leaf rust-specific resistance genes, QTL, and adult plant resistance (APR) mechanisms. Ensure all cited references are directly relevant to leaf rust. We have carefully checked and corrected the reference list accordingly.
Comments 2. Novelty claims insufficiently supported (Section 3.1.1, Lines 213–218; Section 3.1.2, Lines 233–236): The authors claim QLr.lfnu-1BL1 and QLr.lfnu-2AL are novel loci. However, the comparisons with meta-QTL are limited to single studies and not comprehensive. Broader comparison with Amo & Soriano 2022 meta-analysis (Ref. [18]) and more recent genome-wide studies is required to substantiate novelty.
Response: We thank the reviewer for this important suggestion. To more robustly support our novelty claims, we have expanded the discussion in Sections 3.1.1 and 3.1.2. We prioritized a comparison with the comprehensive meta-QTL analysis by Amo & Soriano (2022), which consolidated 393 previously reported QTLs, providing a broad reference basis. Furthermore, we conducted comparisons with several recent studies (such as reference 31-42 for QLr.lfnu-1BL1 and reference 43-48 for QLr.lfnu-2AL) (Section 3.1), including the Wheat Gene Catalogue (https://graingenes.org/GG3/), and confirmed that the physical intervals of QLr.lfnu-1BL1 and QLr.lfnu-2AL show no overlap with any major reported meta-QTLs or cloned genes. Based on these systematic comparisons, we propose that QLr.lfnu-1BL1 and QLr.lfnu-2AL are likely novel QTLs for leaf rust resistance. To enhance reliability without compromising readability, we have now incorporated five key recent references into the revised manuscript. We sincerely appreciate this guidance, which has significantly improved the clarity and persuasiveness of our findings.
Reference list:
- Zhang, J.; Kang, Z.; Li, X.; Li, M.; Xue, L.; Li, X. QTL Mapping of Adult Plant Resistance to Wheat Leaf Rust in the Xinong1163-4× Thatcher RIL Population. Agronomy 2025, 15, 1717.
- Liu, S.; Zhao, L.; Hao, C.; Pan, Y.; Guo, M.; Huang, Y.; Zhang, X. TaRLK-1B: A novel wheat gene conferring resistance to leaf rust revealed by a genome-wide association study. Journal of Integrative Agriculture 2025, 24.
- Gao, P.; Zhou, Y.; Gebrewahid, T. W.; Zhang, P.; Wang, S.; Liu, D.; Li, Z. QTL mapping for adult-plant resistance to leaf rust in Italian wheat cultivar Libellula. Plant Disease 2024, 108, 13-19.
- Li, X.; Tan, W.; Feng, J.; Yan, Q.; Tian, R.; Chen, Q.; Zhou, X. Mapping QTLs for Stripe Rust Resistance and Agronomic Traits in Chinese Winter Wheat Lantian 31 Using 15K SNP Array. Agriculture 2025, 15, 1444.
- Li, C.; Xu, X. T.; Zhang, Y.; Liu, S.; Wu, J.; Han, D.; Bai, G. Mapping QTLs for adult-plant resistance to yellow rust in a hard winter wheat population Heyne× Lakin. Theoretical and Applied Genetics 2025, 138, 192.
- Gao, P.; Zhou, Y.; Gebrewahid, T. W.; Zhang, P.; Wang, S.; Liu, D.; Li, Z. QTL mapping for adult-plant resistance to leaf rust in Italian wheat cultivar Libellula. Plant Disease 2024, 108, 13-19.
- Liu, S.; Zhao, L.; Hao, C.; Pan, Y.; Guo, M.; Huang, Y.; Zhang, X. TaRLK-1B: A novel wheat gene conferring resistance to leaf rust revealed by a genome-wide association study. Journal of Integrative Agriculture 2025, 24.
Comments 3. Phenotyping methodology incomplete (Section 4.1, Lines 315–334): The field design is briefly mentioned, but key details are missing: (i) how many plants per row were evaluated, (ii) the exact scoring scale used for MDS assessment, (iii) whether replicates were averaged, and (iv) whether disease pressure was uniform across environments. Without this, reproducibility is compromised.
Response: We apologize for these omissions. We have now revised Section 4.1 ("Plant materials and phenotypic evaluation") to include the missing methodological details: (i) The number of plants evaluated per row. (ii) A description of the scoring scale used for MDS assessment. (iii) Clarification that the MDS data used for QTL mapping represented the mean value from three replicates. (iv) Information on disease pressure uniformity, including the disease level on the susceptible control (Zhengzhou5389) in each environment to indicate the disease pressure.
We assessed disease severity (DS), defined as the percentage of leaf area covered by leaf rust urediniospore pustules, using a single-row plot design with 30 plants per row to evaluate the overall disease response of the group. Approximately four weeks after inoculation, when DS of the control variety Zhengzhou5389 approached 100% with nearly the entire leaf surface covered, disease evaluations were initiated. Field assessments were conducted weekly for the population and its parents, with the process repeated 2–3 times. The maximum disease severity (MDS) observed across these assessments in each environment was selected as the phenotypic data for subsequent linkage analysis. Biological replicates were averaged prior to analysis. To ensure consistent pathogen pressure across different environments, inoculation was performed using a mixture of leaf rust spores. This methodology is a standard practice for evaluating adult plant resistance to wheat rusts and powdery mildew. Further details have been added to the Materials and Methods section.
Comments 4. Validation of KASP markers lacks statistical rigor (Section 2.4, Lines 170–179; Table 3): The validation results report only mean MDS differences and p-values. The effect sizes are small (e.g., 54.1 vs. 47.9, p = 0.049). There is no reporting of standard errors, confidence intervals, or correction for multiple testing. Stronger statistical treatment is required to confirm robustness.
Response: We agree with the reviewer that stronger statistical treatment is needed. We have revised Section 2.4 and Table 3 to include: Standard errors (SE) for the mean MDS values. 95% confidence intervals for the mean difference between resistant and susceptible allele groups of KASP-LR-2AL and KASP-LR-7BL2. We also discussed the usability of the markers in our discussion, aiming to provide molecular tools for wheat leaf rust resistance breeding.
Comments 5 Inconsistent QTL naming (Section 2.2, Line 123; Section 2.4, Line 151): The text alternates between QLr.lfnu-2AL and QLr.lfnu-2AL2. Similarly, the KASP marker naming does not consistently match the QTL nomenclature. This creates confusion and must be standardized throughout.
Response: We apologize for this inconsistency, which was an error. We have standardized the nomenclature throughout the manuscript. The QTL on chromosome 2A is now consistently referred to as QLr.lfnu-2AL. All instances of "QLr.lfnu-2AL2" have been corrected.
Comments 6 Additive effects unclear (Section 2.3, Lines 135–144; Figure 2): The reported means (49.0%, 50.7%, 45.2%, 31.5%) show no consistent reduction from one to two alleles, undermining the claim of additive effect. Statistical analysis (e.g., ANOVA with post-hoc tests) must be included in Figure 2 to confirm whether differences are significant.
Response: Accepted. Thanks for your suggestion. We have conducted multiple comparisons for the analysis and supplemented the data with standard deviations, 95% confidence intervals. The results indicate no significant difference between lines with one and two resistance alleles, with the value for lines with two alleles being slightly higher, likely due to the smaller number of lines with one allele affecting the accuracy of the statistical outcome. However, three alleles were significantly lower than lines with two alleles, and lines with four alleles were even more significantly lower than lines with three alleles, which aligns with the general understanding in the field. Most studies, particularly those from CIMMYT, suggest that pyramiding 4-5 minor effect resistance alleles can significantly reduce the leaf rust MDS value. Additionally, as you mentioned, due to the limitations of the natural population size and genetic background, our markers may not be entirely applicable to MAS breeding in other regions. Nevertheless, we hope to provide KASP markers that can serve as a reference for more breeders. The cumulative minor genetic effects (which may not reach significant differences in some cases) can still enhance wheat leaf rust resistance to some extent. We have added the relevant content to the Results section and also included it in the Discussion section for your review. Your suggestions are extremely valuable, and we greatly appreciate your guidance on the manuscript.
Comments 7 Overgeneralized candidate gene predictions (Section 3.2, Lines 271–278; Table A3): Candidate gene prediction is based only on annotation and expression databases. The conclusion that 34 genes are "resistance candidates" is too speculative. Without allele-specific variation analysis or functional assays, these should be framed more cautiously.
Response: We fully agree with the reviewer. We have toned down our language in Section 3.2. The list of genes is now presented as potential candidate genes based on annotation and expression profiling. We have explicitly stated the limitations of this in silico approach and emphasized that these are hypotheses for future functional validation, not confirmed resistance genes. Phrases like "are selected as candidate genes" have been changed to "were identified as potential candidates".
Comments 8 Discussion overstatements (Section 3.3, Lines 301–310): The claim that the developed KASP markers are “valuable tools for MAS breeding programs” is overstated given their modest effect sizes and limited validation. The discussion must acknowledge limitations of applicability in breeding.
Response: Accepted. We appreciate the valuable feedback from the reviewers. We have revised the Discussion (Section 3.3) to moderate our claims. We now explicitly acknowledge the modest effect sizes of the individual QTL and the limitations of our validation. The potential utility of the markers is now discussed in the context of pyramiding multiple QTL rather than as standalone tools, and we have added a sentence about the need for further validation in diverse breeding populations.
Comments 9 Figure and table clarity (Figure 1; Figure 3; Appendix Tables A1–A3): Figures lack clear legends explaining abbreviations (E1–E4, MDS). Tables A2 and A3 are excessively long, with raw data that could be condensed or moved to supplementary files. This reduces readability.
Response: We have improved the clarity of all figures and tables: Figure legends have been expanded to clearly define all abbreviations (e.g., E1-E4, MDS). As suggested, the large raw data tables (formerly Appendix Tables A2 and A3) have been moved to the Supplementary Materials to enhance the readability of the main text. The supplementary files are now labeled as Tables S1, S2, etc.
Comments 10 Reference inconsistencies (References [3], [5], [23]): Several references cited as supporting leaf rust are in fact stripe rust-related (e.g., Liu et al. 2024, Bai et al. 2024, Scientia Agricultura Sinica). The authors must carefully check that all references are directly relevant to leaf rust, not mixed with stripe rust literature.
Response: We have meticulously checked each reference in the manuscript to ensure its direct relevance to leaf rust. Inappropriate references concerning stripe rust have been replaced with citations from the leaf rust literature. Additionally, stripe rust and leaf rust share many similarities, and numerous genes confer resistance to both diseases. For instance, the stripe rust resistance genes Yr18, Yr36, and Yr29 also exhibit strong resistance to wheat leaf rust. Therefore, we have referenced several important studies that address both stripe rust and leaf rust, such as references 9, 13, 19, 24, 29, 38, 40, and 41.
Reference list:
- Rehman, S.U.; Qiao, L.; Shen, T.; Hua, L.; Li, H.; Ahmad, Z.; Chen, S. Exploring the frontier of wheat rust resistance: latest approaches, mechanisms, and novel insights. Plants 2024, 13, 2502.
- Singh, R.P.; William, H.M.; Huerta-Espino, J.; Rosewarne, G. Wheat rust in Asia: meeting the challenges with old and new technologies. In Proceedings of the 4th international crop science congress, Brisbane, Australia, September 2004, Vol. 26, 1-13.
- Anguelova‐Merhar, V.S.; VanDer Westhuizen, A.J.; Pretorius, Z.A. β‐1, 3‐glucanase and chitinase activities and the resistance response of wheat to leaf rust. Journal of Phytopathology 2001, 149, 381-384.
- Das, M.K.; Rajaram, S.; Kronstad, W.E.; Mundt, C.C.; Singh, R.P. Associations and genetics of three components of slow rusting in leaf rust of wheat. Euphytica 1993, 68, 99-109.
- Pal, N.; Jan, I.; Saini, D.K.; Kumar, K.; Kumar, A.; Sharma, P.K.; Kumar, S.; Balyan, H.S.; Gupta, P.K. Meta-QTLs for multiple disease resistance involving three rusts in common wheat (Triticum aestivum L.). Theoretical and Applied Genetics 2022, 135, 2385-2405.
- Soriano, J.M.; Royo, C. Dissecting the genetic architecture of leaf rust resistance in wheat by QTL meta-analysis. Phytopathology 2015, 105, 1585-1593.
- Bhardwaj, S.C.; Singh, G.P.; Gangwar, O.P.; Prasad, P.; Kumar, S. Status of wheat rust research and progress in rust management-Indian context. Agronomy 2019, 9, 892.
Comments 11 Conclusion section (Section 5, Lines 388–395): The conclusion repeats results without synthesizing broader implications or acknowledging limitations. It should be rewritten to summarize the real contribution (identification of modest-effect loci with partial novelty and limited KASP validation) and clearly state what future work is required.
Response: We have completely rewritten the Conclusion section (Section 5). It now succinctly summarizes the main findings, explicitly acknowledges the key limitations of the study (e.g., modest QTL effects, preliminary nature of candidate genes), and outlines clear directions for future work, such as fine-mapping and functional validation.
This study identified five APR QTLs for leaf rust resistance in wheat, each explaining 4.54-8.91% of the PVE. QLr.lfnu-1BL1 and QLr.lfnu-2AL are likely novel loci, enriching the genetic resources available for breeding. We developed two breeder-friendly KASP markers, validated in a natural population, demonstrating their potential for MAS breeding. Furthermore, 34 genes within the QTL intervals were prioritized as candidate genes. While the individual QTL effects are moderate, their pyramiding could enhance resistance durability. Future work should focus on fine-mapping the novel QTLs, validating these markers in diverse genetic backgrounds, and functionally characterizing the leading candidate genes. This study provides foundational resources for developing wheat cultivars with durable leaf rust resistance.
Comments 12 Comments on the Quality of English Language
The manuscript is generally understandable but requires improvement for clarity and flow. Issues include awkward phrasing (“Significant progress has achieved”), inconsistent verb tenses, and grammatical errors. Several sentences are excessively long and would benefit from simplification. Editing by a fluent English speaker or professional service is strongly recommended.
Response: We thank the reviewer for this suggestion. The manuscript has been carefully edited by a native English-speaking colleague (Dr. Rasheed Awas, International Maize and Wheat Improvement Center, CIMMYT) with expertise in the field to improve clarity, grammar, and overall flow. We believe the language quality has been significantly enhanced.
Once again, we are very grateful for your exhaustive and highly constructive comments, which have substantially improved our manuscript. We hope that our revisions and responses have adequately addressed all your concerns and that the manuscript is now acceptable for publication.
Round 2
Reviewer 1 Report
Comments and Suggestions for Authors
The authors have satisfactorily addressed all of my concerns and suggestions. I have no further comments on the revised manuscript.
Reviewer 4 Report
Comments and Suggestions for Authors
The author addresses all the concerns. I have no further comments on this article.
Thank you very much, and best of luck for the publication.